# Microscopic origins of the large piezoelectricity of leadfree (Ba,Ca)(Zr,Ti)O$_3$

Yousra Nahas[1], Alireza Akbarzadeh[1], Sergei Prokhorenko[1], Sergey Prosandeev[1,2], Raymond Walter[1], Igor Kornev[3], Jorge Íñiguez[4] & L. Bellaiche[1]

In light of directives around the world to eliminate toxic materials in various technologies, finding lead-free materials with high piezoelectric responses constitutes an important current scientific goal. As such, the recent discovery of a large electromechanical conversion near room temperature in $(1-x)\mathrm{Ba(Zr_{0.2}Ti_{0.8})O_3} - x\mathrm{(Ba_{0.7}Ca_{0.3})TiO_3}$ compounds has directed attention to understanding its origin. Here, we report the development of a large-scale atomistic scheme providing a microscopic insight into this technologically promising material. We find that its high piezoelectricity originates from the existence of large fluctuations of polarization in the orthorhombic state arising from the combination of a flat free-energy landscape, a fragmented local structure, and the narrow temperature window around room temperature at which this orthorhombic phase is the equilibrium state. In addition to deepening the current knowledge on piezoelectricity, these findings have the potential to guide the design of other lead-free materials with large electromechanical responses.

[1] Physics Department and Institute for Nanoscience and Engineering, University of Arkansas, Fayetteville, Arkansas 72701, USA. [2] Research Institute of Physics, Southern Federal University, Rostov on Don 344090, Russia. [3] Laboratoire Structures, Propriétés et Modélisation des Solides, Université Paris-Saclay, CentraleSupélec, CNRS-UMR8580, Grande Voie des Vignes, 92295 Châtenay-Malabry Cedex, France. [4] Department of Materials Research and Technology, Luxembourg Institute of Science and Technology, (LIST), 5 avenue des Hauts-Fourneaux, L-4362 Esch/Alzette, Luxembourg. Correspondence and requests for materials should be addressed to Y.N. (email: yousra.nahas@gmail.com).

Piezoelectricity is a physical phenomenon that converts mechanical into electrical energy and vice-versa (see ref. 1 and references therein). It has been used for devices such as sensors, actuators or ultrasonic motors[2,3]. Up to now, the materials exhibiting the highest piezoelectric responses include lead ions, such as $Pb(Zr,Ti)O_3$ or $Pb(Mg,Nb,Ti)O_3$ (refs 4–8), that introduce toxicity concerns. As a result, the search for lead-free compounds exhibiting large room-temperature piezoelectricity constitutes an important current research direction that is partially driven by regulations announced by several countries[9,10]. In that regard, the discovery of a large electromechanical response found at room temperature in the $(1 - x)Ba(Zr_{0.2}Ti_{0.8})O_3 - x(Ba_{0.7}Ca_{0.3})TiO_3$ solid solutions with $x = 0.50$ (to be denoted as BCTZ-0.5 in the following) and reported in ref. 11 is a major finding.

Interestingly, the microscopic origin of such large piezoelectricity in this lead-free system remains subject to debate. For instance, the experimental study of ref. 11 suggests that it arises from the proximity of a tricritical point, where two ferroelectric phases of rhombohedral and tetragonal symmetries meet with the paraelectric phase of cubic symmetry. In contrast, the combined theoretical and experimental investigation of ref. 12 proposes that the highest piezoelectric coefficients are reached at the boundary between ferroelectric phases of orthorhombic and tetragonal symmetries as a result of a combination of reduced anisotropy energy, high polarization and enhanced elastic softening. The experimental analyses of refs 13,14 offer yet another explanation, pointing out the coexistence of tetragonal, orthorhombic and rhombohedral phases and the strong electric-field-dependency of their relative contributions to the total system as the culprits responsible for the large observed electromechanical response. On the basis of the polarization-rotation mechanism proposed in lead-based materials[4,15–18], it is also legitimate to wonder if an overlooked low-symmetry phase, inside which the spontaneous polarization easily rotates, may be responsible for large piezoelectricity in $(Ba_{0.85}Ca_{0.15})(Zr_{0.10}Ti_{0.90})O_3$.

A plausible explanation for the paucity of knowledge of BCTZ-0.5 is that atomistic simulations, which have been particularly important to understand piezoelectricity in lead-based materials[15–18], are currently lacking for $(1 - x)Ba(Zr_{0.2}Ti_{0.8})O_3 - x(Ba_{0.7}Ca_{0.3})TiO_3$ compounds. This lack of simulations is likely due to the difficulty in realistically mimicking these latter solid solutions, since not only do they possess chemical mixing at both the A and B sublattices of the $ABO_3$ perovskite structure, but can also exhibit local inhomogeneities (especially if phase coexistence occurs as advocated in refs 13,14). As a result, large supercells are most likely required to accurately model BCTZ-0.5, which is typically problematic from the standpoint of memory and computational time.

Here, we build a large-scale atomistic approach to tackle room-temperature piezoelectricity in $(Ba_{0.85}Ca_{0.15})(Zr_{0.10}Ti_{0.90})O_3$. The use of such a scheme leads to a successful explanation of its origin that we find residing in the dyadic combination of the narrow temperature range of stability of the macroscopic orthorhombic phase near 300 K, and the flatness of the free energy associated with this orthorhombic phase, which allows large fluctuations of the polarization around its equilibrium value. Such macroscopic effects are also associated with specific characteristics of the local structures, including the existence of the so-called percolating cluster (which is of orthorhombic symmetry) while its strength (that is, volume per cent occupied in the material) is found here to directly correlate with piezoelectricity. It is also worth realizing that clusters of orthorhombic symmetry are not ingredients of the widely used Comes–Guinier–Lambert[19], which implies that this model ought to be generalized to be more realistic.

## Results

**Atomistic scheme.** We adopt the virtual crystal approximation (VCA)[20,21] mimic $(1 - x)Ba(Zr_{0.2}Ti_{0.8})O_3 - x(Ba_{0.7}Ca_{0.3})TiO_3$ with $x = 0.50$. We first model a virtual $\langle A \rangle \langle B \rangle O_3$ simple perovskite system, for which the $\langle A \rangle$ atom involves a compositional average of Ba and Ca potentials of 85 and 15% respective contributions, while the $\langle B \rangle$ atom is built from a mixing of the Zr and Ti potentials of 10 and 90% respective contributions. An effective Hamiltonian ($H_{eff}$) is then developed for this $\langle A \rangle \langle B \rangle O_3$ system. As in ref. 22, the degrees of freedom of this $H_{eff}$ are, for each 5-atom unit cell $i$, the local soft mode $\mathbf{u}_i$ that is proportional to the electric dipole moment of the cell, the $\eta_H$ homogeneous strain tensor, and inhomogeneous-strain-related dimensionless displacements $\{\mathbf{v}_i\}$. Technically, the various $\{\mathbf{u}_i\}$ and $\{\mathbf{v}_i\}$ are, respectively, centred on the $\langle B \rangle$ and $\langle A \rangle$ sites. The analytical expression for the total internal energy of this effective Hamiltonian is the one provided in ref. 22 for pure $BaTiO_3$, and therefore contains a local-mode self-energy, a long-range dipole–dipole interaction, a short-range interaction between soft modes, an elastic energy, and an interaction between the local modes and local strains. In particular, the local-mode self-energy is given by:

$$E_{tot} = \sum_i \left\{ \kappa_2 u_i^2 + \alpha u_i^4 + \gamma \left( u_{i,x}^2 u_{i,y}^2 + u_{i,y}^2 u_{i,z}^2 + u_{i,z}^2 u_{i,x}^2 \right) \right\}, \quad (1)$$

where the sum runs over all the $\langle B \rangle$ sites and where ($u_{i,x}$, $u_{i,y}$, $u_{i,z}$) are the Cartesian components of $\mathbf{u}_i$ in the orthonormal basis formed by the [100], [010] and [001] pseudo-cubic directions. In the first step, the parameters $\kappa_2$, $\alpha$ and $\gamma$ are determined, along with the other coefficients of the effective Hamiltonian, by performing density functional theory calculations within the VCA approach[21] on small $\langle A \rangle \langle B \rangle O_3$ cells (less than 20 atoms). In a second step, Monte-Carlo (MC) simulations using $E_{tot}$ are conducted on large supercells (typically of $12 \times 12 \times 12$ or $18 \times 18 \times 18$ dimensions) made of $\langle A \rangle \langle B \rangle O_3$. During these simulations, $\kappa_2$ is varied to fit the experimental value of the Curie temperature[11,12,23] (since $H_{eff}$ techniques can underestimate the paraelectric–ferroelectric transition temperature[22,24]) and $\gamma$ is slightly adjusted to reproduce the measured lowest transition temperature observed in refs 12,23 for $(1 - x)Ba(Zr_{0.2}Ti_{0.8})O_3 - x(Ba_{0.7}Ca_{0.3})TiO_3$ solid solutions having $x = 0.50$. For comparison, we also computed finite-temperature properties of pure $BaTiO_3$ (BTO), as arising from the use of the effective Hamiltonian of ref. 24.

**Phase transitions.** The results of these MC simulations for BCTZ-0.5 and BTO are shown in Fig. 1a,b for the Cartesian components of the supercell average of the local modes, $\langle \mathbf{u} \rangle$ (which is directly proportional to the spontaneous polarization), when averaging over 4 million MC sweeps and using $18 \times 18 \times 18$ supercells. The $x$, $y$ and $z$ axes are chosen along the [100], [010] and [001] pseudo-cubic directions, respectively. The computations for BCTZ-0.5 correctly qualitatively and even quantitatively reproduce the three observed successive (first-order) transitions[12,23] when cooling down the system: first, a paraelectric cubic $Pm\bar{3}m$ to ferroelectric tetragonal $P4mm$ transition at around 360 K, for which the $z$ component of $\langle \mathbf{u} \rangle$ becomes nonzero; second, a $P4mm$ to ferroelectric orthorhombic $Amm2$ transition near 297 K (that is, very close to room temperature), for which the $y$ component of the supercell average of the local modes suddenly becomes nonzero and equal to the $z$ component; and third, a $Amm2$ to ferroelectric rhombohedral $R3m$ transition at $\simeq 270$ K, for which all the Cartesian components of $\langle \mathbf{u} \rangle$ are now nonzero and equal to each other. In Supplementary Note 2, it is also demsonstrated that the effective Hamiltonian scheme used within the VCA approach (with a rescaling of the $\kappa_2$ and $\gamma$ parameters) can correctly reproduce the temperature–compositional phase diagram of

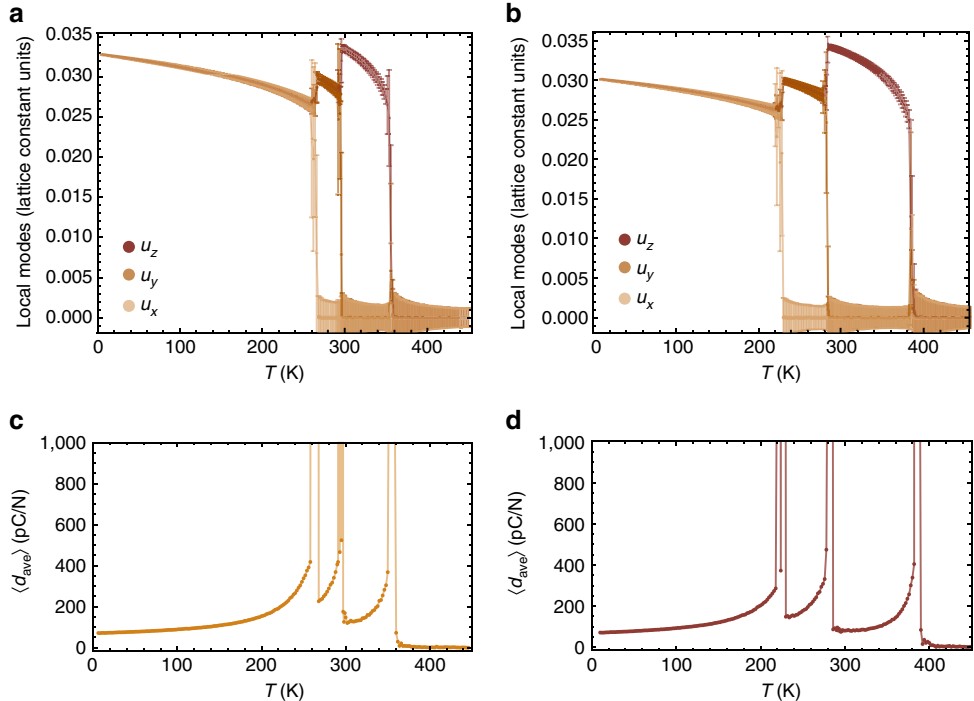

**Figure 1 | Temperature dependence of macroscopic properties. (a,b)** Report the evolution with temperature $T$ of the supercell average of the local modes (in lattice constant units) of $(1-x)\mathrm{Ba(Zr_{0.2}Ti_{0.8})O_3} - x\mathrm{(Ba_{0.7}Ca_{0.3})TiO_3}$ with $x = 0.50$ (BCTZ-0.5) and $\mathrm{BaTiO_3}$ (BTO), respectively. Error bars indicate s.d., and are less than or equal to the size of the points when not visible. **(c,d)** Show the evolution with temperature $T$ of the average $\langle d_{ave}\rangle$ piezoelectric coefficient (see text) for BCTZ-0.5 and BTO, respectively. Note that $\langle d_{ave}\rangle$ reaches values larger than 1,000 at the $Pm\bar{3}m - P4mm$, $P4mm - Amm2$ and $Amm2 - R3m$ transitions. Such values are not shown here because they will make it challenging to see piezoelectricity of the order of hundreds of pC/N in **c,d**. Results in all panels are obtained from the use of $18 \times 18 \times 18$ supercells.

$(1-x)\mathrm{Ba(Zr_{0.2}Ti_{0.8})O_3} - x\mathrm{(Ba_{0.7}Ca_{0.3})TiO_3}$ (BCTZ-$x$) for $x$ ranging between 0.25 and 0.65, as well as other properties of BCTZ-$x$, which further attests to the validity of the presently used numerical method. Moreover, for pure $\mathrm{BaTiO_3}$, Fig. 1b shows that the $Pm\bar{3}m - P4mm$, $P4mm - Amm2$ and $Amm2 - R3m$ transitions are predicted to be about 384, 283 and 226 K, which agree reasonably well with the corresponding experimental values of 400, 280 and 180 K (refs 25,26).

**Piezoelectricity.** Piezoelectric coefficients, $d_{ij}$, of BCTZ-0.5 and BTO are calculated using the correlation-function approach of ref. 27, that is:

$$d_{ij} = \frac{N_s Z^* a}{k_B T}\left\{\left\langle u_i \eta_{H,j}\right\rangle - \left\langle u_i\right\rangle\left\langle\eta_{H,j}\right\rangle\right\}, \quad (2)$$

where $T$ is the temperature, $k_B$ the Boltzmann constant, $N_s$ the total number of 5-atom cells composing the supercell, $Z^*$ the Born effective charge associated with the soft mode and $a$ the 5-atom lattice constant. $u_i$ is the $i$-component of the supercell average of the local mode at a given MC sweep, and $\eta_{H,j}$ is the $j$ component of the homogeneous strain tensor (in Voigt notation) at this MC sweep. The $\langle\rangle$ symbol denotes statistical averages over the different MC sweeps. Figure 1c,d report an averaged computed piezoelectric coefficient, $\langle d_{ave}\rangle$, for BCTZ-0.5 and $\mathrm{BaTiO_3}$, respectively. More precisely, $\langle d_{ave}\rangle$ is equal to $\frac{d_{11} + d_{22} + d_{33}}{3}$ (for the aforementioned $(x, y, z)$ basis) in the R3m phase, since $d_{11}$, $d_{22}$ and $d_{33}$ coefficients are all nonzero and equal to each other in this phase. On the other hand, $\langle d_{ave}\rangle$ is chosen to be $\frac{d_{11} + d_{22} + d_{33}}{2}$ (respectively, $d_{11} + d_{22} + d_{33}$) in the $Amm2$ ($P4mm$) phase because only two (one) of these three coefficients are (is) nonzero there. We also practically choose $\langle d_{ave}\rangle$ to be equal to

$d_{11} + d_{22} + d_{33}$ in the paraelectric phase, as in the tetragonal ferroelectric state (note that all three aforementioned choices provide the same $\langle d_{ave}\rangle$ in the $Pm\bar{3}m$ phase since it has no piezoelectricity).

In BCTZ-0.5, large values of this piezoelectric coefficient exist in the temperature range associated with the stability of the $Amm2$ phase, as consistent with the experimental findings of refs 11,12. In particular, the computed averaged $\langle d_{ave}\rangle$ coefficient is always bigger than $\simeq 225$ pC/N and can be as high as 525 pC/N in the macroscopic $Amm2$ phase of BCTZ-0.5 (note that piezoelectric coefficients larger than 525 pC/N shown in Fig. 1c correspond to frequent fluctuations between different macroscopic phases, such as $Amm2$ and $P4mm$, and are therefore inherently linked to phase transitions). In contrast, the piezoelectric coefficient can be as small as $\simeq 150$ pC/N and does not exceed values of about $\simeq 330$ pc/N in the macroscopic $Amm2$ state of $\mathrm{BaTiO_3}$. Interestingly and unlike in lead-based $\mathrm{Pb(Zr,Ti)O_3}$ and $\mathrm{Pb(Mg,Nb,Ti)O_3}$ solid solutions near their morphotropic phase boundary[4,15,16], these large piezoelectric responses in BCTZ-0.5 are not due to the existence of a low-symmetry (that is, monoclinic) phase that is associated with the ease of rotating the polarization, since our calculations reported in Fig. 1a (as well as corresponding data related to strain tensors that are not shown here) indicate that they occur within a macroscopic orthorhombic phase.

It is also worthwhile to realize that Fig. 1c further predicts that BCTZ-0.5 should also have large piezoelectric coefficients (larger than 200 pC/N) in the rhombohedral R3m phase in the vicinity of the $Amm2$–R3m phase transition, namely for temperature varying between 260 and 266 K, as well as close to the Curie temperature of $\simeq 360$ K, which is consistent with the experimental data of ref. 12.

**Fluctuations**. To better understand the piezoelectric responses within the $Amm2$ phases of BCTZ-0.5 and BTO, as well as their differences and origins, Fig. 2a reports $\langle d_{ave} \rangle$ versus $\delta u_y^2 + \delta u_z^2$ in the ferroelectric orthorhombic state of these two materials, where $\delta u_y$ and $\delta u_z$ are the error bars that are associated with the $y$ and $z$ components of the supercell average of the local mode that are shown in Fig. 1a,b. These error bars quantify the fluctuation of the polarization about its macroscopic average, which is oriented along the [011] direction. What is remarkable is that not only $\langle d_{ave} \rangle$ is found to nearly linearly increase with $\delta u_y^2 + \delta u_z^2$, but also that this linear relationship is rather similar between BCTZ-0.5 and BaTiO$_3$ (Supplementary Note 3 provides an understanding of the linear relationship between $\langle d_{ave} \rangle$ and $\delta u_y^2 + \delta u_z^2$. We did not include $\delta u_x$ in the computation of the fluctuations reported in the horizontal axis of Fig. 2a because the polarization in the orthorhombic phase has a vanishing $x$ component). One can thus assert that BCTZ-0.5, in contrast with BTO, can adopt values of $\langle d_{ave} \rangle$ larger than 330 pc/N in its $Amm2$ state because it can sustain larger fluctuations of its polarization. Moreover, unlike BaTiO$_3$, BCTZ-0.5 is prevented from having piezoelectric coefficients smaller than 225 pC/N because the narrow temperature stability of its macroscopic $Amm2$ phase (about 30 K in BCTZ-0.5 versus 60 K in BTO, see Fig. 1a,b) prevents the orthorhombic symmetry from reaching lower temperatures where thermal fluctuations are restricted. Note that the inherent relation between a limited temperature range of stability of the

orthorhombic phase and large piezoelectricity is further demonstrated in the Supplementary Note 2, where we also examined BCTZ-0.4. As a matter of fact, in this latter system, the range of stability of its orthorhombic state further contracted (about 15 K), which results in even larger piezoelectric coefficients. However, such range of stability in BCTZ-0.4 occurs for temperatures higher than 300 K, which is detrimental to generate high room-temperature electromechanical response.

To understand the larger fluctuations occurring in the orthorhombic phase of BCTZ-0.5 with respect to the case of BaTiO$_3$, Fig. 3 displays a quantity related to free energy–internal energy of both systems at their $P4mm$–$Amm2$ transition, as obtained using the Wang-Landau algorithm of ref. 28 within the effective Hamiltonians presently developed and/or used here. More precisely, this quantity related to free energy corresponds to the logarithm of the canonical probability function[28]. Figure 3 demonstrates the existence of two minima of similar free energy in both systems. The right minimum with larger internal energies corresponds to the macroscopic $P4mm$ state while the left minimum with smaller internal energies is associated with $Amm2$. The existence of these two minima demonstrates the first-order character of the $P4mm$–$Amm2$ transition. Figure 3 also indicates that the energetic barrier between these two minima is smaller in BCTZ-0.5 than in BTO, therefore making the exploration of different orthorhombic states of different polarization direction, via an intermediate tetragonal state, easier of access close to the $P4mm$–$Amm2$ transition in $(Ba_{0.85}Ca_{0.15})(Zr_{0.10}Ti_{0.90})O_3$ than in BaTiO$_3$. This finding is fully consistent with the suggestion of ref. 11 that BCTZ-0.5 possesses a low energetic barrier between different ferroelectric states that allows its polarization to easily rotate and that results in large piezoelectric responses very near the $P4mm$–$Amm2$ transition. Furthermore and understand Figs 2a and 3 also reveals that the free energy–internal energy curve is much flatter around the orthorhombic minimum in BCTZ-0.5 than in BTO. In other words, there is a wider range of orthorhombic states having different internal energies (and thus different magnitudes of the polarization) that hold a similar free-energy in BCTZ-0.5. As a result, BCTZ-0.5 can exhibit larger fluctuations of its polarization within the macroscopic $Amm2$ phase. Note also that the existence of a low free-energy barrier and of flat minima revealed in Fig. 3 for BCTZ-0.5 is consistent with the

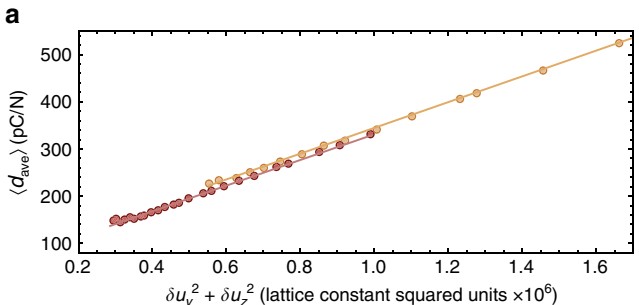

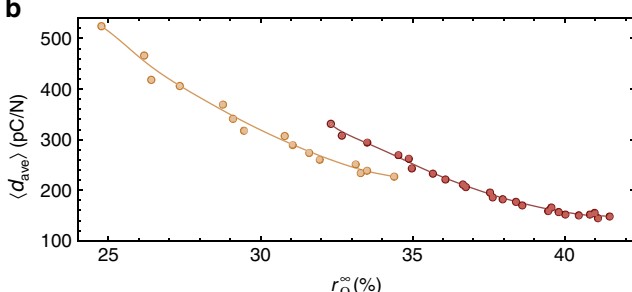

**Figure 2 | Dependence of piezoelectricity on fluctuations and percolation.** (**a**) Reports the dependency of the average $\langle d_{ave} \rangle$ piezoelectric coefficient (see text) on fluctuations of the polarization in the macroscopic $Amm2$ phase of $(1-x)Ba(Zr_{0.2}Ti_{0.8})O_3 - x(Ba_{0.7}Ca_{0.3})TiO_3$ with $x = 0.50$ (BCTZ-0.5) in yellow symbols, and BaTiO$_3$ (BTO) in red symbols. $\delta u_y$ and $\delta u_z$ are the error bars of the $y$ and $z$ component of the supercell average of the local mode, respectively, displayed in Fig. 1a,b. (**b**) Shows the dependency of the average $\langle d_{ave} \rangle$ piezoelectric coefficient on the $r_O^\infty$ strength of the percolating O cluster (that is, the infinite cluster that spreads from one side of the supercell to its opposite side along the [100], [010] or [001] pseudo-cubic directions[36]) in the macroscopic $Amm2$ phase of BCTZ-0.5 (yellow symbols) and BTO (red symbols). Results correspond to the use of a $18 \times 18 \times 18$ supercell. Solid lines are linear least-square fits in **a**, and guide for the eyes in **b**.

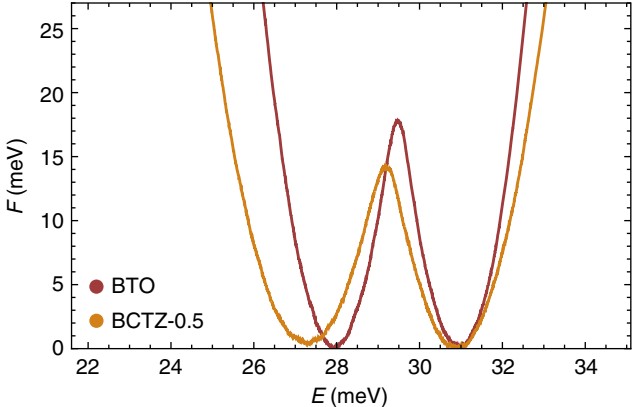

**Figure 3 | Free-energy landscape.** Free-energy-like quantity versus internal energy (in meV per 5-atom cell) for $(1-x)Ba(Zr_{0.2}Ti_{0.8})O_3 - x(Ba_{0.7}Ca_{0.3})TiO_3$ with $x = 0.50$ (BCTZ-0.5) in yellow, and BaTiO$_3$ (BTO) in red, at their $P4mm$–$Amm2$ transition temperatures (which are 297 and 283 K, respectively). Results correspond to the use of a $18 \times 18 \times 18$ supercell. This free-energy-like quantity is the logarithm of the normalized canonical distribution[28].

proximity of a tricritical point in the phase diagram of $(1-x)Ba(Zr_{0.2}Ti_{0.8})O_3 - x(Ba_{0.7}Ca_{0.3})TiO_3$, as discussed in ref. 11 (see Supplementary Note 2 for our predictions and discussion about tricritical point in this solid solution).

**Cluster analysis**. Let us now determine whether the piezoelectric response and polarization's fluctuations of BCTZ-0.5 and BTO correlate with microscopic features. For that, we identified clusters of tetragonal (T), Orthorhombic (O) and Rhombohedral (R) symmetry (within which dipoles nearly all lie along a $\langle 001 \rangle$, $\langle 110 \rangle$ and $\langle 111 \rangle$ pseudo-cubic direction, respectively) in both BCTZ-0.5 and BaTiO$_3$, using a modified version of the Hoshen–Kopelman algorithm[29,30]. Interestingly, we numerically found (not shown here) that both systems support T, O and R clusters in their macroscopic $P4mm$ state; this is reminiscent of the coexistence of $P4mm$, $Amm2$ and $R3m$ phases close to the $P4mm$–$Amm2$ transition, reported in ref. 13 based on a Rietveld analysis of X-ray powder diffraction for a $(Ba_{0.85}Ca_{0.15})(Zr_{0.10}Ti_{0.90})O_3$ sample. Moreover, while our T clusters fully vanish in the $Amm2$ state of these two materials, the O and R clusters remain, which bears resemblance with the decrease of the per cent of $P4mm$ phase experimentally reported in refs 13,14 when subjecting BCTZ-0.5 to electric field or stress near room temperature. These changes in microstructures are associated with the huge piezoelectric response found at the $P4mm$–$Amm2$ transition occurring between 296 and 298 K. Furthermore, our findings about local clusters demonstrate that the microscopic and macroscopic symmetries of BaTiO$_3$, but also of BCTZ-0.5, can be quite different, which is in line with the celebrated Comes–Guinier–Lambert model[19]. On the other hand, our predictions of the existence of T clusters (in the $P4mm$ phases) and O clusters (in both the $P4mm$ and $Amm2$ phases) in addition to R clusters are in line with the findings of ref. 31 and go beyond the Comes–Guinier–Lambert model, since this latter model only expects R clusters (with different $\langle 111 \rangle$ directions) to occur in the macroscopic $P4mm$ and $Amm2$ phases.

Let us focus on the macroscopic orthorhombic $Amm2$ phase of both BCTZ-0.5 and BTO since we are interested in relating its large piezoelectric coefficients displayed in Fig. 1c with atomistic characteristics. In this phase, we numerically found that the R clusters are dynamical in nature[32], since they can change of location within the supercell and can also jump from one $\langle 111 \rangle$ direction to another $\langle 111 \rangle$ direction between different MC sweeps. Note that jump of polarization is typically associated with the so-called central mode, as demonstrated in ref. 33 for pure BaTiO$_3$. On the other hand, we further discovered that there are two different types of O clusters in the macroscopic orthorhombic $Amm2$ phase of BCTZ-0.5 and BaTiO$_3$. One type has a strong dynamical character as a result of the different $\langle 110 \rangle$ directions and locations within the supercell it can adopt during the MC simulations at fixed temperature. On the other hand, the second type of O clusters has a pronounced static character in the sense that its polarization is always oriented along the spontaneous polarization. However, this second type of cluster also possesses some dynamics, that is, it breathes rather than change location during these simulations, which may be related to the soft-mode that has been predicted and observed to exist (in addition to the central mode) in BTO[33,34]. As a result, such second type of O cluster can be referred to as quasi-static (it is worthwhile to realize that our simulations thus show that both quasi-static and dynamical clusters can coexist inside a pure system, such as BaTiO$_3$, and not only in complex solid solutions such as relaxor ferroelectrics[35]). Interestingly, this quasi-static type of O cluster is, in fact, the so-called percolating cluster that

spreads from one side of the supercell to its opposite side along the [100], [010] or [001] pseudo-cubic directions[36].

To corroborate these observations, we have conducted additional molecular dynamics simulations so as to estimate the relative time-scale of cluster dynamics in BCTZ-0.5 within the orthorhombic phase, at 280 K. We found that the polarization of the percolating O cluster does not change orientation throughout the 400 ps total simulation time, thus being indeed quasi-static at these time scales accessible to molecular dynamics simulations. Moreover, we found that the autocorrelation time of the volume of the non-percolating O clusters is of $\sim 1.4$ ps, while for the percolating O cluster, it is of $\sim 2.3$ ps, hence indicating that the latter, while featuring significantly slower dynamics nevertheless has a breathing component to its time evolution, essentially due to volume fluctuations. Examples of R clusters, as well as the two types of O clusters, are shown in Fig. 4e,f for both BCTZ-0.5 and BaTiO$_3$. Figure 4a–d indicate the relative evolution with temperature of different types of clusters in BCTZ-0.5 and BaTiO$_3$, and show that the priorly evidenced enhancement of piezoelectric response in the $Amm2$ phase of the former is associated with a more fragmented local structure. Specifically, Fig. 4a reports the ratio between the number of sites belonging to all R clusters over the total number of sites in the whole supercell in the macroscopic orthorhombic $Amm2$ phase of these two systems and as a function of temperature, while Fig. 4b displays the same ratio but for all O clusters. These two ratios are denoted as $r_R$ and $r_O$, respectively. Figure 4c shows the so-called strength of the percolating O cluster, $r_O^\infty$, corresponding to the per cent of sites belonging to the (infinite) percolating O cluster. Furthermore, Fig. 4d reports the difference between $r_O$ and $r_O^\infty$, that is it represents the per cent of total volume occupied by the aforementioned first type of O clusters (that is, by the dynamical O clusters). Figure 4a,b demonstrate that the R and O clusters occupy a significant amount of the whole supercell in the macroscopic $Amm2$ state of both BCTZ-0.5 and BTO. For instance, for BaTiO$_3$ at 240 K, $r_R$ and $r_O$ are both close to 40%. Interestingly, comparing Fig. 4b–d also tells us that most of the space occupied by the O clusters originates from the percolating O cluster, as demonstrated by the fact that $r_O–r_O^\infty$ is always smaller than $\simeq 6\%$ for any temperature and decreases down to $\simeq 1\%$ when decreasing the temperature to 230 K. Other important information provided by Fig. 4 is that $r_O$ and thus $r_O^\infty$ are rather sensitive to temperature in the $Amm2$ state of both BCTZ-0.5 and BTO, unlike $r_R$. For instance, $r_O^\infty$ increases from 25 to 41% when decreasing temperature from 296 to 230 K, while $r_R$ remains close to 40% in both the studied materials in that temperature range. Recalling that the range of stability of the $Amm2$ state typically occurs for higher temperatures in BCTZ-0.5 than in BaTiO$_3$, one can therefore conclude that the local structure of the $Amm2$ phase of BCTZ-0.5 is more disordered/fragmented than that of BaTiO$_3$. Such enhanced disordering allows for easier fluctuations of the polarization, and thus according to Fig. 2a to larger piezoelectricity. Figure 2b confirms the correlation between large piezoelectricity and enhancement of disordering of the local structure, as well as further sheds light into the strong connection between large electromechanical responses and percolating clusters, since $\langle d_{ave} \rangle$ is found to typically increase when $r_O^\infty$ decreases in both BCTZ-0.5 and BTO.

To confirm this observation, we have additionally estimated the contribution to piezoelectricity stemming from each type of clusters in the $Amm2$ phase of BCTZ-0.5 (at 280 K) and BTO (at 250 K), by first determining at each MC sweep which local modes belong to which type of clusters and then using equation (2) for the local modes associated with each type of clusters. We found that, in the case of BCTZ-0.5 (BTO), the percolating O cluster, which occupies 31% (38.3%) of the

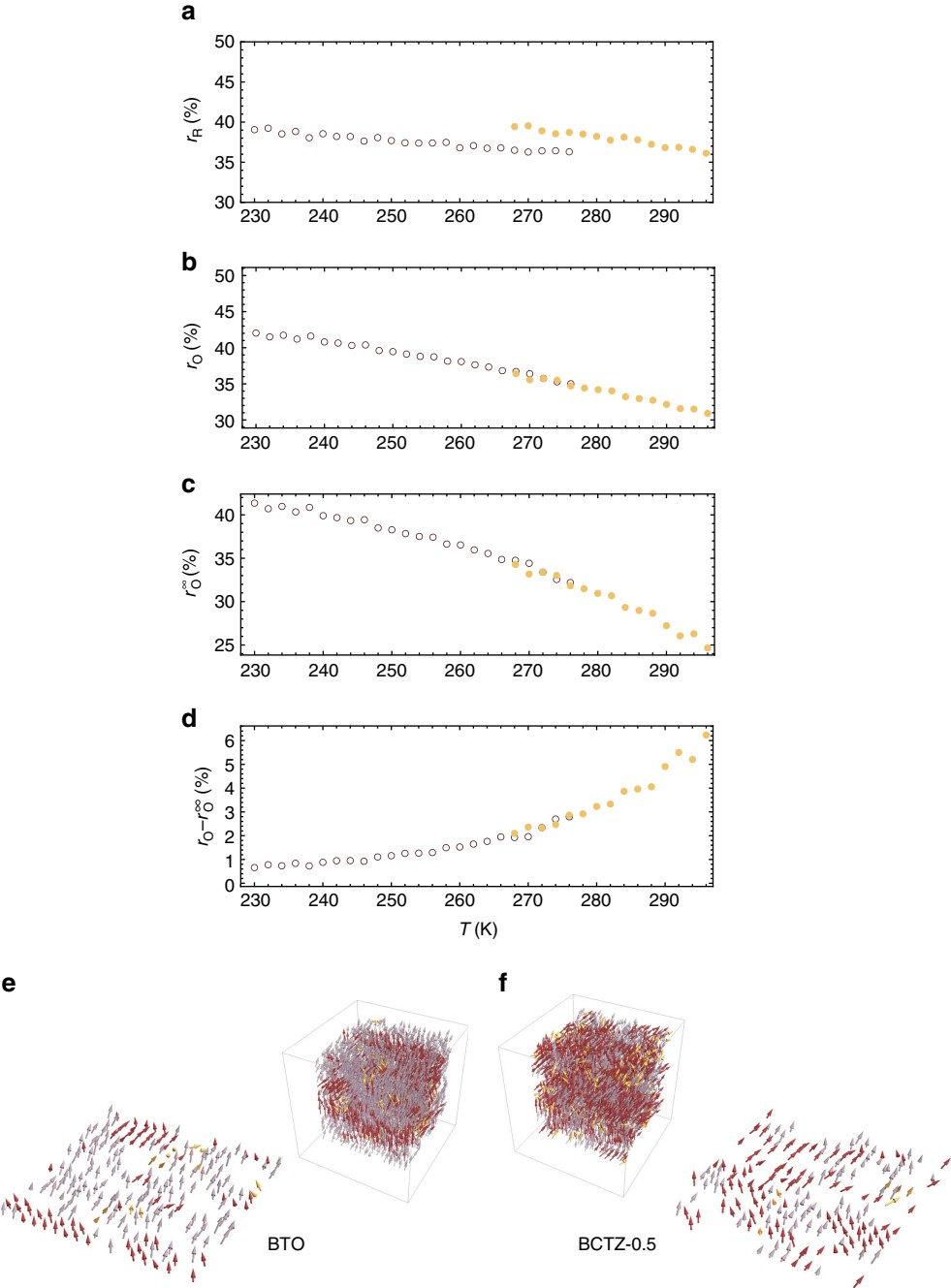

**Figure 4 | Cluster analysis.** (**a**,**b**) Display the percentage occupied by the R and O clusters within the volume of the supercell ($r_R$ and $r_O$). (**c**) Represents the $r_O^\infty$ strength of the percolating O cluster (that is, the infinite cluster that spreads from one side of the supercell to its opposite side along the [100], [010] or [001] pseudo-cubic directions[36]). (**d**) Represents the difference between $r_O$ and $r_O^\infty$. These data (**a**–**d**) are shown for BCTZ-0.5 (filled symbols) and BTO (open symbols) in their macroscopic *Amm*2 state, and correspond to the average over 100 different dipolar configurations (associated with 100 different MC sweeps) at any considered temperature *T* in a 18 × 18 × 18 supercell. (**e**,**f**) Display examples of R clusters (in red) as well as the two types of O clusters (the non-percolating in yellow, and percolating ones in purple) as occurring within the supercell and in a given *xy* supercell cross-section in BTO at 250 K and in BCTZ-0.5 at 280 K, respectively.

supercell, has an individual piezoelectric response of 70.5 pC/N (48.9 pC/N), while the dynamical R and non-percolating O clusters, occupying, respectively, 38.4% (37.8%) and 3.3% (1.2%) of the supercell, have piezoelectric contributions of 68.8 pC/N (27.3 pC/N) and 29.5 pC/N (11.5 pC/N). These results consistently indicate that the larger the volume of the percolating O cluster, the lower is its contribution to the piezoelectric response (see Supplementary Note 1 for additional information about BCTZ-0.5). In light of these results, the trend line to achieve

enhanced piezoelectricty appears to rest upon the relative fragmentation of local order. The latter can be tuned via *x* (see Supplementary Note 2 for additional information about BCTZ-*x*), via the application of a small electric field along one of the equivalent ⟨111⟩ directions that would depopulate the percolating O cluster, or alternatively, given the interplay between epitaxial strain and the orientation and morphology of local order[37], via the application of epitaxial strain. Note, however, that these levers that would allow the tuning of the ratio of R and non-percolating

O clusters to percolating O clusters in favour of the former ones are in interplay with temperature, a parameter that is intrinsically related to the studied phenomenon via thermal fluctuations.

## Discussion

In summary, atomistic simulations within an effective Hamiltonian scheme predict that BCTZ-0.5 undergoes a $P4mm$–$Amm2$ transition that occurs near room temperature, and that yields an orthorhombic state that has a rather flat free-energy landscape as well as a small temperature range of stability. A result, larger fluctuations of the polarization occur in the $Amm2$ state of BCTZ-0.5 with respect to BaTiO$_3$, thereby inducing higher piezoelectric responses near 300 K. Moreover, our study further reveals that this larger piezoelectric response is intrinsically linked to a specific feature of the local structure, namely the smaller strength of the percolating cluster. Interestingly, such cluster is of orthorhombic rather than rhombohedral local symmetry, and, as result, is missing in the famous Comes–Guinier–Lambert model[19]. In other words, such latter model ought to be generalized (by including O clusters in the $Amm2$ phase) be able to capture the microscopic origins of physical properties of BCTZ-0.5 and BTO.

Note that our study focuses on single phases. However, we also expect that the formation and coexistence of several phases inside BCTZ-0.5 will contribute to further enhancing piezoelectricity. This expectation stems from the fact that the barrier height of the free energy corresponds to the interface tension or, in other words, to the energy of the domain wall between phases of different symmetry[38], and, from this perspective, the reduction of the barrier height revealed in Fig. 3 when going from BaTiO$_3$ to BCTZ-0.5 enables domain wall fluctuations that should further strengthen the electromechanical response[39].

We hope that such findings not only provide a better understanding of BCTZ-$x$ systems but also can be used in the quest for other lead-free systems with high electromechanical conversion. For instance, our results suggest that one possibility for generating large piezoelectricity is to mix one system having a single-phase transition from cubic paraelectric to ferroelectric rhombohedral at a temperature to be denoted by $T_{c1}$, with another material having another single-phase transition but from cubic paraelectric to ferroelectric tetragonal at a temperature to be denoted $T_{c2}$. This mixing can then result in the emergence of a ferroelectric orthorhombic state having flat free-energy minimum, within a narrow temperature region that is located in-between the temperature stability regions of the ferroelectric tetragonal and ferroelectric rhombohedral states. The key factor to then obtain large room-temperature piezoelectric response is to make the stability of the orthorhombic state occurring in a region comprising room temperature, which can happen by finding the right amount of mixing between the two compounds and the right $T_{c1}$ and $T_{c2}$ temperatures. Note that, in this scenario, the existence of a tricritical point, where the tetragonal and rhombohedral ferroelectric states meet with the paraelectric phase, can occur at a specific composition and temperature but the highest room-temperature piezoelectric response can correspond to other compositions, namely the ones for which the orthorhombic state is stable within a small temperature region comprising 280–300 K. These are precisely the conditions encountered in BCTZ-$x$ (refs 12,23). Note also that this scenario is different from, that is, the mixing of rhombohedral Pb(Zr,Ti)O$_3$ of lower Ti compositions with tetragonal Pb(Zr,Ti)O$_3$ of larger Ti concentrations for which the coexistence of rhombohedral and tetragonal domains[40], or the occurrence of a compositionally induced bridging monoclinic phase[4,15], can yield large electromechanical response, since our scenario consists in creating a macroscopic orthorhombic phase of small temperature range of stability in-between the temperature ranges of the tetragonal and rhombohedral states.

**Data availbility.** The data that support the findings of this study are available from the corresponding author upon reasonable request.

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

## Acknowledgements

We thank Drs R. Ranjan and J. Kreisel for insightful discussions. Y.N. and L.B. acknowledge the ARO grant W911NF-16-1-0227. A.A. and S.Prok. thank the DARPA grant HR0011-15-2-0038 (MATRIX programme). S.Pros. appreciates the ONR Grant N00014-12-1-1034 and grants 3.1649.2017/PP from RME (Russian Ministry of Education) and 16-52-0072 Bel_a from RFBR (Russian Foundation for Basic Research). R.W. thanks the National Science Foundation Graduate Research Fellowship Program under Grant No. DGE-0957325 and the University of Arkansas Graduate School Distinguished Doctoral Fellowship. We also acknowledge the support of the Luxembourg National Research Fund through the PEARL (Grant FNR/P12/4853155/ Kreisel COFERMAT, J.Í.) and inter-mobility (Grant FNR/INTER/MOBILITY/15/ 9890527 GREENOX, L.B. and J.Í.) programs. S.Pros. also appreciates Russian Ministry of Education grant 3.1649.2017/PP.

## Author contributions

Y.N. performed effective Hamiltonian simulations and conducted Wang-Landau calculations, cluster and fluctuation analysis. A.A. computed the phase diagram displayed in the Supplementary Fig. 4. S.Prok. and Y.N. performed molecular dynamics simulations for assessing clusters dynamics, and computed their individual piezoelectric responses. S.Pros., R.W. and L.B. extracted the effective Hamiltonian parameters. I.K. developed the Wang-Landau code that was used to obtain Fig. 3, and provided discussions along with J.Í., S.Prosandeev. and S.Prokhorenko. L.B. wrote the original version of the manuscript, which was then extensively modified due to the feedback and suggestions of all authors.

## Additional information

**Competing interests:** The authors declare no competing financial interests.

**Publisher's note**: 

DOI: 10.1038/ncomms16172    **OPEN**

# Corrigendum: Microscopic origins of the large piezoelectricity of leadfree (Ba,Ca)(Zr,Ti)O$_3$

Yousra Nahas, Alireza Akbarzadeh, Sergei Prokhorenko, Sergey Prosandeev, Raymond Walter, Igor Kornev, Jorge Íñiguez & L. Bellaiche

*Nature Communications* 8:15944 doi:10.1038/ncomms15944 (2017); Published 20 Jun 2017; Updated 25 Oct 2017

The affiliation details for Sergei Prokhorenko are incorrect in this Article. A second affiliation should have been included as given below:

Theoretical Materials Physics, Q-MAT CESAM, Université de Liège, B-4000 Sart Tilman, Belgium.

Also, the financial support for this Article was not fully acknowledged. The Acknowledgements should have included the following:

Sergei Prokhorenko thank the University of Liege and the EU in the context of the FP7-PEOPLE-COFUND-BeIPDproject.

