## [Peer Review File · Nature Communications]

Reviewers' comments:

Reviewer #1 (Remarks to the Author):

The present manuscript describes an atomic scale investigation of the origin of the very large piezoelectric response in (Ba,Ca)(Ti,Zr)O₃ ceramics that was experimentally observed several years ago. To this end, the authors adopt an advanced and elegant simulation approach that was developed and refined by some of them over the course of many years, and adapt it to the system at hand. In this fashion they provide a convincing and insightful analysis of the microscopic features that lead to the remarkable macroscopic features of this material observed experimentally.

Overall I think this a very good paper that is well written and provides an assessment that is accessible to both experimentalists and theoreticians in the field. I appreciate the compactness of the argumentation and also the conclusions/discussion section, which clarifies the difference in the mechanism for high piezoelectricity identified here with respect to e.g., the one in PZT.

There are a couple of detailed comments (see below) the authors should address prior to publication, after which manuscript can probably be recommended for publication.

Detailed comments:

- 1) How much does the critical value of the Curie temperature and the lowest transition temperature change during the adjustment of kappa and gamma? How much do the parameters change? This information should be included in the Supplementary Material.
- 2) The discussion/description of the clusters at the bottom of page 12/top of page 13 is the least readable part of the paper. While one can parse the information, I think a slight rewrite with more emphasis on the content (rather than the references to various Figure subpanels) would improve readability. In the same context, I think a cross section of the cluster distribution shown in Figure 4 would be more illuminating for the wider audience. In such a figure one could also more easily (as a reader) identify the different types of clusters including the differentiation between percolating (quasi-static) and dynamic clusters.
- 3) I find the connection drawn at the bottom of page 13 between experimental synthesis/growth conditions and the form of local structure observed in the present simulations very tenuous. The present simulations do not account for chemical disorder, which is quite probably a factor in the experiments and are not bound to a timescale either. Yet chemical inhomogeneities have been related to polar nanoregions in e.g., relaxors. (Maybe a point worthwhile commenting on, even if only in the Supplementary Material). Hence, I would rather avoid this point or substantiate the claim.
- 4) Fig. 3: Please adopt more widely used energy units, eV or similar.
- 5) Why do the d values in Figs. 2a and b reach different maximum levels?
- 6) The Supplementary Material was only available as a LaTeX file, which was very inconvenient for review, in particular since the connection to the figures cited in the text was unclear.

Reviewer #2 (Remarks to the Author):

I think that while this is a good paper, it should be made stronger to warrant publication in Nature Communications. The authors investigate the BCTZ material demonstrated to have high piezoelectric performance in 2009. The authors show that the piezoelectric coefficient d is

proportional to the fluctuations of the polarization, which in turn are large due to the narrow temperature stability range of the room-temperature orthorhombic phase and to the flatness of the potential energy surface. The authors also demonstrate that at the microscopic level, these features are associated with a combination of R and O clusters.

I think that the linear relationship between the polarization fluctuation and the d_{ij} parameter is rather obvious and expected. Similarly, the fact that a narrow temperature stability range of a ferroelectric phase favors a flat potential energy surface and that such a flat potential energy surface give rise to large fluctuation is also not unexpected from basic Landau theory arguments. While the quantitative evaluation of these phenomena for the BCTZ system is valuable, I don't think that it is suitable in and of itself for publication in Nature Communications.

On the other hand, the authors show the interesting local structure of a combination of R and O polar clusters with a mixture of dynamic and static behavior. However, the connection between this combination and the dielectric and piezoelectric response of BCTZ is rather qualitative. Therefore, I think that the paper would be greatly improved and much more novel if the authors would quantitatively characterize the contributions of the R and O clusters to the dielectric and piezoelectric response and perhaps relate these to the local distributions of the Ba/Ca and Ti/Zr cations as was done in their previous work on the BZT relaxors. For example, it would be interesting to see whether there is a difference in electromechanical coupling constants of the R, O (static) and O (dynamic) clusters. Such a revised paper could be suitable for publication in Nature Communications.

I also would like to comment on a technical point. The authors make a distinction between static and dynamics O clusters. The static polarization of the cluster is possibly only static on the simulation time scale. Since the authors used the MC approach, can they estimate the time scale for which such static behaviour is observed? In other words, it is possible that the static clusters also migrate and their polarization orientation changes in the similar way to the dynamic clusters, but on a much longer time scale that is not probed in the simulations, but would be probed experimentally, where the time scale is several orders of magnitudes greater. Therefore, it would be useful to provide an estimate of the timescale examined in the MC simulations.

Reviewer #3 (Remarks to the Author):

The paper reports on the results and interpretation of a large scale atomistic model of the piezoelectric material $(\text{Ba,Ca})(\text{Zr,Ti})\text{O}_3$ using a Virtual Crystal Approximation approach. The results are interesting to those in the field in that they indicate that the high room temperature piezoelectric coefficients are consequence of the narrowness of the $\text{Amm}2$ phase in the temperature domain and hence the proximity of the structural phase transitions to the $\text{P}4\text{mm}$ and $\text{R}3\text{c}$ phases. The role of the normal softening of the lattice in the proximity of the transitions, which is always a major contributor to the coefficient is somewhat under-played by the authors. They prefer to highlight the role of heterophase fluctuations in enhancing the properties. Nevertheless, the work is important as it shows that the previously cited proximity to the triple point is not so relevant.

The authors have possibly carried their analysis too far. Is the detail of the analysis in pages 12 and 13 entirely warranted, especially for a more general audience, given that the VCA approach is after all an "approximation" and in many practitioners' eyes a flawed one when the outputs are

examined in such fine detail.

Whilst I believe that the findings are important enough to publish in NComms, I would like to see a more realistic comparative analysis of Figs 1(c) and 1(d) in which the critical softening of the lattice on approaching the Amm2-P4mm transition and the influence of the relatively high temperature of the R3c-Amm2 transition in BCZT is taken into account, before the influence of fluctuations is added in. I recommend this as an optional modification only.

RESPONSE TO REVIEWER 1:

(o.) *The present manuscript describes an atomic scale investigation of the origin of the very large piezoelectric response in (Ba,Ca)(Ti,Zr)O₃ ceramics that was experimentally observed several years ago. To this end, the authors adopt an advanced and elegant simulation approach that was developed and refined by some of them over the course of many years, and adapt it to the system at hand. In this fashion they provide a convincing and insightful analysis of the microscopic features that lead to the remarkable macroscopic features of this material observed experimentally. Overall I think this a very good paper that is well written and provides an assessment that is accessible to both experimentalists and theoreticians in the field. I appreciate the compactness of the argumentation and also the conclusions/discussion section, which clarifies the difference in the mechanism for high piezoelectricity identified here with respect to e.g., the one in PZT.*

Authors' response: We thank Reviewer 1 for these comments and for emphasizing some of our main results, e.g. “a convincing and insightful analysis of the microscopic features that lead to the remarkable macroscopic features of this material observed experimentally” and “which clarifies the difference in the mechanism for high piezoelectricity identified here with respect to e.g., the one in PZT”.

(1.) *How much does the critical value of the Curie temperature and the lowest transition temperature change during the adjustment of kappa and gamma? How much do the parameters change? This information should be included in the Supplementary Material.*

Authors' response: As requested, this information is now included in the Supplementary Material.

(2.) *The discussion/description of the clusters at the bottom of page 12/top of page 13 is the least readable part of the paper. While one can parse the information, I think a slight rewrite with more emphasis on the content (rather than the references to various Figure subpanels) would improve readability. In the same context, I think a cross section of the cluster distribution shown in Figure 4 would be more illuminating for the wider audience. In such a figure one could also more easily (as a reader) identify the different types of clusters including the differentiation between percolating (quasi-static) and dynamic clusters.*

Authors' response: We thank Reviewer 1 for these suggestions, and accordingly add a sentence prior to the description of different panels composing Fig. 4 that emphasizes its main message. With respect to the visualization, we also follow the advice of Reviewer 1 and provide a cross-sectional view of the cluster distribution (see our new version of Figure 4) to render this figure more readable.

(3.) *I find the connection drawn at the bottom of page 13 between experimental synthesis/growth conditions and the form of local structure observed in the present simulations very tenuous. The present simulations do not account for chemical disorder, which is quite probably a factor in the experiments and are not bound to a timescale either. Yet chemical inhomogenities have been related to polar nanoregions in e.g., relaxors. (Maybe a point worthwhile commenting on, even if only in the Supplementary Material). Hence, I would rather avoid this point or substantiate the claim.*

Authors' response: This is a valid point, especially since different growth conditions can result in different arrangements of the alloyed Ba/Ca and/or Ti/Zr elements and that we do not explicitly account for chemical disorder here. As a result, we removed this point in the new version of the manuscript.

(4.) *Fig. 3: Please adopt more widely used energy units, eV or similar.*

Authors' response: We now provide a revised figure 3 displaying the data in meV.

(5.) *Why do the d values in Figs. 2a and b reach different maximum levels?*

Authors' response: Thank you for pointing this out. We had inadvertently omitted three data points within the O-phase of BCTZ-0.5 in the vicinity of the O/T transition in Fig.2 (b). We have now added missing data points in a revised version of figure 2.

(6.) *The Supplementary Material was only available as a LaTeX file, which was very inconvenient for review, in particular since the connection to the figures cited in the text was unclear.*

Authors' response: We apologize for this inconvenience, and now provide the Supplemental Material as a pdf file.

RESPONSE TO REVIEWER 2:

(1.) *I think that while this is a good paper, it should be made stronger to warrant publication in Nature Communications. The authors investigate the BCTZ material demonstrated to have high piezoelectric performance in 2009. The authors show that the piezoelectric coefficient d is proportional to the fluctuations of the polarization, which in turn are large due to the narrow temperature stability range of the room-temperature orthorhombic phase and to the flatness of the potential energy surface. The authors also demonstrate that at the microscopic level, these features are associated with a combination of R and O clusters.*

Authors' response: We thank Reviewer 2 for stating that this is a good paper and for summarizing some of our results. Please note that there are also other findings that the other two Reviewers viewed as important, namely that our work “clarifies the difference in the mechanism for high piezoelectricity identified here with respect to e.g., the one in PZT” (see comment “o” of Reviewer 1) and “the previously cited proximity to the triple point is not so relevant” (see comment 1 of Reviewer 3).

(2.) *I think that the linear relationship between the polarization fluctuation and the d_{ij} parameter is rather obvious and expected. Similarly, the fact that a narrow temperature stability range of a ferroelectric phase favors a flat potential energy surface and that such a flat potential energy surface give rise to large fluctuation is also not unexpected from basic Landau theory arguments. While the quantitative evaluation of these phenomena for the BCTZ system is valuable, I don't think that it is suitable in and of itself for publication in Nature Communications.*

Authors' response: Indeed, the connection between polarization fluctuation and the piezoelectric coefficient is usually intuitively expected. Note, however, that we derived (based on simple Landau theory arguments) and provide in Eqs. (5) and (6) of the Supplemental material the precise form of this relation, in general, and indicate that it involves a fluctuation of the square of some (but not all) components of the electric dipoles, in particular. We thus hope that this derivation will be useful to the scientific community, since we do not recall to have seen it before. Regarding the flatness of the potential energy surface and the resulting large fluctuation of the polarization, it is indeed typically expected in the scientific community but, to the best of our knowledge, never proven in atomistic simulations because it, e.g, requires the access to the free energy at finite temperature. We are thus glad to see that our newly developed Wang-Landau scheme incorporated within our atomistic effective Hamiltonian is able to find such flat potential energy surface and that it is accompanied by large polarization fluctuations. Such numerical findings are promising for atomistic schemes since they further increase the capabilities and predictive power of such schemes, in our humble opinion.

(3.) *On the other hand, the authors show the interesting local structure of a combination of R and O polar clusters with a mixture of dynamic and static behavior. However, the connection between this combination and the dielectric and piezoelectric response of BCTZ is rather qualitative. Therefore, I think that the paper would be greatly improved and much more novel if the authors would quantitatively characterize the contributions of the R and O clusters to the dielectric and piezoelectric response and perhaps relate these to the local distributions of the Ba/Ca and Ti/Zr cations as was done in their previous work on the BZT relaxors. For example, it would be interesting to see whether there is a difference in electromechanical coupling constants of the R, O (static) and O (dynamic) clusters. Such a revised paper could be suitable for publication in Nature Communications.*

Authors' response: We thank the Reviewer for this interesting suggestion about relative contributions of clusters to piezoelectric and dielectric responses (note however that we can not correlate with the local distributions of Ba/Ca and Ti/Zr ions because we use the VCA approach which does not explicitly account for chemical disorder). Accordingly, we have computed the contributions of each type of clusters, namely, R clusters, percolating O cluster (static), and non-percolating O clusters (dynamic), to the dielectric and piezoelectric responses in the case of BCTZ-0.5 at 280 K, i.e., within the O-phase, where they occupy 38.35%, 31%, and 3.25% of the supercell, respectively. We found the following:

- * *Piezoelectric response:* the largest contribution stems from the percolating O cluster (70.5 pC/N), followed by the R clusters (68.8 pC/N), and the non-percolating O clusters (29.5 pC/N). Results further indicate that the weaker

contribution of the non-percolating O clusters to the piezoelectric response results from smaller correlation of their polarization with strain due to the dynamic behavior of dipoles within these clusters. Moreover, we find that the component of polarization in R clusters that is orthogonal to the supercell orthorhombic polarization has a dynamic behavior associated with large value of the corresponding cumulant. For comparison, we have also estimated the contributions stemming from various clusters in BaTiO₃ at 250 K, i.e., within the O-phase, where they occupy 37.8%, 38.3%, and 1.2% of the supercell, for R clusters, percolating O cluster, and non-percolating O clusters, respectively. We found that the largest contribution to the piezoelectric response is once again due to the percolating O cluster (48.9 pC/N), followed by the R clusters (27.3 pC/N), and the non-percolating O clusters (11.5 pC/N). Note that the comparison of these results between BCTZ-0.5 at 280 K and BaTiO₃ at 250 K also demonstrates that the larger the volume of the percolating O cluster, the lower is its individual piezoelectric response.

- * *Dielectric response:* the largest contribution stems from R clusters (equal to 551.5 for BCTZ-0.5 at 280 K), followed by the percolating O cluster (equal to 296.3 for BCTZ-0.5 at 280 K), and the non-percolating O clusters (165.30 for BCTZ-0.5 at 280 K). These results show that, although the R clusters and the percolating O cluster occupy comparable volumes in the supercell, the contribution of the R cluster is almost 2 times bigger than that of the percolating cluster, due to the dynamic behavior of the former. Similarly, despite the non-percolating clusters having almost ten times smaller volume compared to that occupied by the percolating O cluster, their contribution to the dielectric response is only twice lower, compensation brought by their dynamic behavior.

(4.) *I also would like to comment on a technical point. The authors make a distinction between static and dynamics O clusters. The static polarization of the cluster is possibly only static on the simulation time scale. Since the authors used the MC approach, can they estimate the time scale for which such static behaviour is observed? In other words, it is possible that the static clusters also migrate and their polarization orientation changes in the similar way to the dynamic clusters, but on a much longer time scale that is not probed in the simulations, but would be probed experimentally, where the time scale is several orders of magnitudes greater. Therefore, it would be useful to provide an estimate of the timescale examined in the MC simulations.*

Authors' response: In order to address this interesting point, we have conducted additional molecular dynamics simulations so as to estimate the relative time-scale of cluster dynamics in BCTZ-0.5 within the orthorhombic phase, at 280 K. We found that the polarization of the percolating O cluster does not change orientation through out the 400 ps total simulation time, thus being quasi-static at these time scales accessible to molecular dynamics simulations. Moreover, we found that the autocorrelation time of the volume of the non-percolating O clusters is of ~ 1.4 ps, while for the percolating O cluster, it is of ~ 2.3 ps, hence indicating that the latter, while featuring slower dynamics nevertheless has a breathing component to its time evolution, essentially due to volume fluctuations.

RESPONSE TO REVIEWER 3:

(1.) *The paper reports on the results and interpretation of a large scale atomistic model of the piezoelectric material (Ba,Ca)(Zr,Ti)O₃ using a Virtual Crystal Approximation approach. The results are interesting to those in the field in that they indicate that the high room temperature piezoelectric coefficients are consequence of the narrowness of the Amm2 phase in the temperature domain and hence the proximity of the structural phase transitions to the P4mm and R3c phases. The role of the normal softening of the lattice in the proximity of the transitions, which is always a major contributor to the coefficient is somewhat under-played by the authors. They prefer to highlight the role of heterophase fluctuations in enhancing the properties. Nevertheless, the work is important as it shows that the previously cited proximity to the triple point is not so relevant.*

Authors' response: We thank this Reviewer for emphasizing some of our important findings, namely that (i) our results “indicate that the high room temperature piezoelectric coefficients are consequence of the narrowness of the Amm2 phase in the temperature domain and hence the proximity of the structural phase transitions to the P4mm and R3c phase”; and (ii) “the previously cited proximity to the triple point is not so relevant”.

Regarding the softening of the lattice in the proximity of the transitions, please note that Figure 1 of the Supplemental Material does report such softening. We are concerned that the fact that we submitted a latex file rather than a pdf file for this Supplemental Material prevented this Reviewer from seeing these data. If this is the case, we apologize for that.

(2.) *The authors have possibly carried their analysis too far. Is the detail of the analysis in pages 12 and 13 entirely warranted, especially for a more general audience, given that the VCA approach is after all an “approximation” and in many practitioners’ eyes a flawed one when the outputs are examined in such fine detail.*

Authors' response: As indicated in our response to point 3 of Reviewer 1, we agree that we have carried our analysis too far when commenting experimental results on piezoelectricity using different growth procedures. We thus removed this point in our new version of the manuscript. On the other hand, we believe that it is important to continue to emphasize the discovered feature of the local structures having different types of clusters, because (i) we also found it in BaTiO₃, implying that this feature does not require the use of the VCA technique for modeling solid solutions to occur (since BaTiO₃ is a simple system); (ii) these findings are novel and important as emphasized by, e.g, the reports of Reviewer 1 (see his/her point “o”) and Reviewer 2 (see his/her third and fourth points) and the fact that this feature automatically implies that the celebrated Comes-Guinier-Lambert model has to be generalized; and (iii) this feature allows to explain on a microscopic level why the (macroscopic) piezoelectric response can be large in BCTZ.

(3.) *Whilst I believe that the findings are important enough to publish in NComms, I would like to see a more realistic comparative analysis of Figs 1(c) and 1(d) in which the critical softening of the lattice on approaching the Amm2-P4mm transition and the influence of the relatively high temperature of the R3c-Amm2 transition in BCZT is taken into account, before the influence of fluctuations is added in. I recommend this as an optional modification only.*

Authors' response: We thank this Reviewer for stating that “I believe that the findings are important enough to publish in NComms”. Please note that we provided some information about lattice softening versus temperature in the Supplemental Material (see Fig. S1). Please also note that our atomistic simulations automatically include fluctuations of the degrees of freedom of the effective Hamiltonian (i.e., local modes and strains) because of the nature of the Monte-Carlo algorithm. Such feature therefore prevents us from investigating the critical softening of the lattice on approaching the Amm2-P4mm transition and the influence of the relatively high temperature of the R3c-Amm2 transition in BCZT without these fluctuations. We are also concerned that neglecting these fluctuations will significantly change some important physical quantities, such as the values of the critical transition temperatures and the temperature range of stability of the different phases.

SUMMARY OF CHANGES:

(note that these changes are indicated in red in the manuscript and Supplemental Material)

- In response to point 1 of Reviewer 1, we added a paragraph on pages 5 and 6 of the Supplemental Material. This paragraph reads “Note that the aforementioned rescaling of the κ_2 and γ parameters is needed to have a phase diagram that quantitatively agrees with measurements. For instance, using the initial values of $\kappa_2 = 0.0626$ a.u. and $\gamma = 0.4453$ a.u. (along with a negative pressure of -4.8 GPa, as done in Ref. [7] for pure BaTiO₃) provides for $x=0.50$ critical temperatures of $\simeq 127\text{K}$, 121K and 116K for the cubic-to-tetragonal, tetragonal-to-orthorhombic and orthorhombic-to-rhombohedral transitions, respectively. They are therefore smaller and closer to each other than the corresponding values of $\simeq 360\text{K}$, 297K and 270K obtained when adopting the presently renormalized $\kappa_2 = 0.0383$ a.u. and $\gamma = 0.4186$ a.u. (without applying a negative pressure). The need for such rescaling, in general, and the reduction of κ_2 , in particular (which results in deeper energy wells), may originate from our use of the local approximation within DFT to extract the parameters of the effective Hamiltonian, as argued in a recent work [10]. It may also be due to the virtual crystal approximation by itself.” Within this paragraph, a new reference (namely, reference 10) is added and quoted.
- In response to points 2, 4 and 5 of Reviewer 1, we now provide revised versions of Figures 2, 3 and 4, and add the following sentence “Panels (a) to (d) of Fig. 4 indicate the relative evolution with temperature of different types of clusters in BCTZ-0.5 and BaTiO₃, and show that the priorly evidenced enhancement of piezoelectric response in the Amm2 phase of the former is associated with a more fragmented local structure” just before describing the different panels composing figure 4.
- In response to point 3 of Reviewer 1, we removed the sentence “Interestingly, these findings about correlations between local structure and piezoelectricity also imply that different growth procedures, or qualities, of the BCTZ-0.5 sample should result in rather different values of the piezoelectric response since these procedures and qualities would inevitably yield different microstructures. This fact has actually been experimentally established in Ref. [34] where the use of various calcined and sintering temperatures in the growth technique have resulted in the variation of the d_{33} coefficients of BCTZ-0.5 ceramics from less than 250 pC/N to more than 600 pC/N at room temperature.” initially appearing at the bottom of page 13, just before the *Conclusions* section. For the same reason, we also removed Ref. [34].
- In response to point 3 of Reviewer 2, we have added the following sentence just before the *Conclusions* section: “To confirm this observation, we have additionally estimated the contribution to piezoelectricity stemming from each type of clusters in the Amm2 phase of BCTZ-0.5 (at 280 K) and BTO (at 250 K), by first determining at each MC sweep which local modes belong to which type of clusters and then using equation 2 for the local modes associated with each type of clusters. We found that, in the case of BCTZ-0.5 (BTO), the percolating O cluster, which occupies 31% (38.3%) of the supercell, has an individual piezoelectric response of 70.5 pC/N (48.9 pC/N), while the dynamical R and non-percolating O clusters, occupying respectively 38.4% (37.8%) and 3.3% (1.2%) of the supercell, have piezoelectric contributions of 68.8 pC/N (27.3 pC/N) and 29.5 pC/N (11.5 pC/N). These results consistently indicate that the larger the volume of the percolating O cluster, the lower is its contribution to the piezoelectric response.”. We have also added the following sentences on page 3 of the Supp. Mat.: “We have also computed, using equation 2 of this Supp. Mat., the contributions of each type of clusters, namely, R clusters, percolating O cluster (static), and non-percolating O clusters (dynamic), to the dielectric responses in the case of BCTZ-0.5 at 280 K, i.e., within the O-phase, where they occupy 38.4%, 31%, and 3.3% of the supercell, respectively. We found that the largest contribution stems from R clusters (551.5), followed by the percolating O cluster (296.3), and the non-percolating O clusters (165.3). These results show that, although the R clusters and the percolating O cluster occupy comparable volumes in the

supercell, the contribution of the R cluster is almost 2 times bigger than that of the percolating cluster, due to the dynamic behavior of the former. Similarly, despite the non-percolating clusters having almost ten times smaller volume compared to that occupied by the percolating O cluster, their contribution to the dielectric response is only twice lower, compensation brought by their dynamic behavior. Note however that the full susceptibility also includes term stemming from correlations of polarization between different cluster types that we have not computed.”

- In response to point 4 of Reviewer 2, we added the following sentences on page 12 of the manuscript: “To corroborate these observations, we have conducted additional molecular dynamics simulations so as to estimate the relative time-scale of cluster dynamics in BCTZ-0.5 within the orthorhombic phase, at 280 K. We found that the polarization of the percolating O cluster does not change orientation through out the 400 ps total simulation time, thus being indeed quasi-static at these time scales accessible to molecular dynamics simulations. Moreover, we found that the autocorrelation time of the volume of the non-percolating O clusters is of ~ 1.4 ps, while for the percolating O cluster, it is of ~ 2.3 ps, hence indicating that the latter, while featuring significantly slower dynamics nevertheless has a breathing component to its time evolution, essentially due to volume fluctuations”.

REVIEWERS' COMMENTS:

Reviewer #1 (Remarks to the Author):

The authors have fully addressed my comments from the previous review round and I now consider this manuscript suitable for publication.

Reviewer #2 (Remarks to the Author):

I think the authors have addressed the comments of the referees well and I think that the current version of the paper could be published in Nature Communications. Still, I think the authors can make their paper even stronger by discussing the possible approaches to change the relative distributions of the R, percolating O and non-percolating O cluster. Since the non-percolating O clusters and R clusters show much stronger individual contributions to the piezoelectric response, adjusting the relative fractions of the different clusters to increase the R and non-percolating O content (this should have an especially strong effect for the non-percolating O due to its low content and high individual piezoelectric response) seems to be an obvious approach for enhancing piezoelectricity in these solid solutions. It would be interesting and would add value to the paper if the authors would discuss whether such adjustment of the relative fractions of different cluster types is feasible, and if so, how it could be accomplished.

Reviewer #3 (Remarks to the Author):

I believe that the authors have replied fully to the referees' comments and have amended their manuscript accordingly. I am therefore happy for the paper to be published with these amendments.

RESPONSE TO REVIEWER 1:

The authors have fully addressed my comments from the previous review round and I now consider this manuscript suitable for publication.

Authors' response: We thank Reviewer 1 for his/her feedback that has helped us improve our manuscript and are pleased to see that he/she finds the revised manuscript suitable for publication.

RESPONSE TO REVIEWER 2:

I think the authors have addressed the comments of the referees well and I think that the current version of the paper could be published in Nature Communications. Still, i think the authors can make their paper even stronger by discussing the possible approaches to change the relative distributions of the R, percolating O and non-percolating O cluster. Since the non-percolating O clusters and R clusters show much stronger individual contributions to the piezoelectric response, adjusting the relative fractions of the different clusters to increase the R and non-percolating O content (this should have an especially strong effect for the non-percolating O due to its low content and high individual piezoelectric response) seems to be an obvious approach for enhancing piezoelectricity in these solid solutions. It would be interesting and would add value to the paper if the authors would discuss whether such adjustment of the relative fractions of different cluster types is feasible, and if so, how it could be accomplished.

Authors' response: We thank Reviewer 2 for his/her feedback that has helped us improve our manuscript and are pleased to see that he/she finds the revised manuscript suitable for publication. With respect to the possible enhancement of piezoelectricity in $(1-x)\text{Ba}(\text{Zr}_{0.2}\text{Ti}_{0.8})\text{O}_{3-x}(\text{Ba}_{0.7}\text{Ca}_{0.3})\text{TiO}_3$ via the control of relative fractions of different clusters, it appears to be in direct relation with x . As discussed in Supplementary Note 2 - Additional information for BCTZ- x - where we have examined BCTZ- x with $x=0.4$, tuning x allows to enhance the fragmentation of local order and subsequently increases the fluctuations of polarization and the piezoelectric response. This, however, happens at temperatures greater than 300 K. For this reason we have stressed this point by explicitly stating that "Such latter features allows greater thermal fluctuations and therefore enable an enhancement of piezoelectricity but at temperatures higher than 300 K. In this regard, $(1-x)\text{Ba}(\text{Zr}_{0.2}\text{Ti}_{0.8})\text{O}_{3-x}(\text{Ba}_{0.7}\text{Ca}_{0.3})\text{TiO}_3$ with $x = 0.50$ is an optimal composition for technological applications in that the large piezoelectric response, although not extremal, ideally occurs near room temperature." Another possibility for leveraging on the fragmentation of local order so as to induce increased piezoelectricity appears to be through the application of a small electric field along one of the $\langle 111 \rangle$ directions that would depopulate the percolating O cluster while increasing the fraction of R clusters. Alternatively, given the interplay between epitaxial strain and the orientation and morphology of local order (see, e.g., Properties of Epitaxial Films Made of Relaxor Ferroelectrics by S. Prosandeev, D. Wang and L. Bellaiche, Physical Review Letters, 111, 247602, 2013) one could also apply strain in order to tune the ratio of R and non-percolating O clusters to percolating O cluster in favor of the former ones. We thus add the following sentence just before the Discussion part of the manuscript:

"In light of these results, the trend line to achieve enhanced piezoelectricity appears resting upon the relative fragmentation of local order. Latter can be tuned via x (see Supplementary Note 2 for additional information about BCTZ- x), via the application of a small electric field along one of the equivalent $\langle 111 \rangle$ directions that would depopulate the percolating O cluster, or alternatively, given the interplay between epitaxial strain and the orientation and morphology of local order³⁷, via the application of epitaxial strain. Note, however, that these levers that would allow to tune the ratio of R and non-percolating O clusters to percolating O cluster in favor of the former ones are in interplay with temperature, a parameter that is intrinsically related to the studied phenomenon via thermal fluctuations."

Where Ref. 37 has been added and corresponds to S. Prosandeev, D. Wang and L. Bellaiche, Physical Review Letters, 111, 247602, 2013.

RESPONSE TO REVIEWER 3:

I believe that the authors have replied fully to the referees' comments and have amended their manuscript accordingly. I am therefore happy for the paper to be published with these amendments.

Authors' response: We thank Reviewer 3 for his/her feedback that has helped us improve our manuscript and are pleased to see that he/she finds the revised manuscript suitable for publication.